# Injectable Resin Technique as a Restorative Alternative in a Cleft Lip and Palate Patient: A Case Report

**DOI:** 10.3390/medicina59050849

**Published:** 2023-04-28

**Authors:** Kelly R. V. Villafuerte, Alyssa Teixeira Obeid, Naiara Araújo de Oliveira

**Affiliations:** 1South America Center for Education and Research in Public Health, Universidad Privada Norbert Wiener, Lima 15046, Peru; 2Dental Division, Restorative Dentistry of the Craniofacial Anomalies Rehabilitation Hospital—HRAC, University of São Paulo, Bauru 17012-230, Brazil; 3Department of Dentistry, Endodontics and Dental Materials, University of São Paulo, Bauru 17012-901, Brazil

**Keywords:** composite resin, cleft lip, cleft palate, operative dentistry

## Abstract

*Objective*: The objective of this study is to present a case report in which the injectable composite resin technique was used as a restorative alternative for dental re-anatomization in a patient with cleft lip and palate and aesthetic complaints. *Materials and Methods*: The treatment plan included the re-anatomization of the maxillary premolars and canines using a flowable composite resin. This resin was injected and cured through a transparent matrix, which was a copy of the diagnostic wax-up model. Some parameters such as application time and marginal adaptation were also observed when performing the restorations. Additionally, old composite resin restorations on the upper lateral incisors were replaced using the incremental technique with conventional resins, which helped to assess color stability and fracture/wear deterioration for both restorative techniques. *Results*: The clinical case report shows that the injectable technique was a simple and quick method for restoring the anatomy of teeth (shape and contour) in one session, since the injectable resin can be easily applied in interproximal areas without the need to manually sculpt the resin. In this case, no clinical, visual, or photographic differences were found in marginal discoloration, color stability, and fracture/wear deterioration for the two restorative techniques after one year of follow-up. *Conclusions*: The professional may have another clinical option for restorative treatment in the case of small re-anatomizations. In addition, the injectable technique seems to require less operator skill and chair time and better marginal adaptation in cases of small anatomical changes.

## 1. Introduction

Through nanotechnology, it has become possible to improve dental restorative materials such as composite resins, especially fluid resins [1,2,3]. By providing nanoscale filler particles, it is possible to add a greater volume of filler to the resin matrix, favoring physical–mechanical characteristics [2,4] and improving optical properties [5]. In addition, due to the volumetric increase in filler content, the resin presents better handling, viscosity, and flow characteristics [6], offering good adaptation to the internal wall of the cavity, thus allowing its use in anterior and posterior restorative procedures [7].

Studies [8,9] indicate that high-volume fluid resins offer better polishing and shine than conventional resins. Thus, the use of this type of resin (or universal injectable) has become common among professionals, leading to the use of various techniques, such as the injectable fluid resin technique. Because it has a simple protocol [1], requires less skill from the operator and less working time, and offers lower cost to patients [10,11], some clinicians opt for indirect treatments due to a lack of skill in using direct resins or in cases of extensive aesthetic restorations [12].

Patients with cleft lip and palate require extensive restorative treatments due to a series of dental anomalies [13]. The cleft lip and palate are caused by the interruption of normal development during the early embryonic period of pregnancy, having a complex phenotype [14]. In addition, the existence of the sub-phenotype was emphasized based on the coexistence of dental anomalies alongside the main phenotypes of cleft, such as unilateral and bilateral cleft lip and palate, isolated palate cleft, or lip cleft [14,15,16]. Facial appearance is central to the psychosocial development of individuals with cleft lip, alveolus, and palate, who often exhibit characteristics of social introversion [17,18]. One of the main reasons may be dental anomalies, which have been shown to be higher in individuals with cleft compared to the normal population [19], including dental agenesis, supernumerary teeth, microdontic upper lateral incisors, the ectopic eruption of teeth, delayed tooth eruption, and enamel hypoplasia, which are frequent in these patients [13,14,20,21]. Additionally, the presence of soft tissue bands connecting the cleft to the base of the nostril or the alveolar margin, known as Simonart’s bands, is common [22,23]. These bands may contribute to a higher prevalence of distal upper lateral incisors to the cleft compared to patients without this condition [23]. Simonart’s band could assist in the relationship in the upper arch, with a better prognosis in treatment; however, it is not the only factor that will guarantee good resolution [23]. Therefore, these conditions result in long treatments carried out in hospitals and specialized places. Although dental treatments in most patients are usually completed between the ages of sixteen to eighteen years old, some require additional aesthetic or functional treatment during adulthood [24].

The injectable fluid resin technique is classified as a minimally invasive direct/indirect technique that preserves tooth structure [2]. In some cases, it does not require prior or dental preparation and can be performed in one or several teeth in the same session [1,2], through analog or digital diagnosis, to optimize aesthetic (shape and contour) and functional (occlusal) parameters [25]. The anatomical shape of the patient’s natural dentition can also be used to create the initial model, which will depend on the type of restorative approach to be selected [1,2]. In addition, this technique can be used in both deciduous teeth [2,26] and permanent teeth [1,2,25,27,28,29], offering various applicability, including applications related to direct veneers [4,25], re-anatomization, and diastema closure [12,27]. However, the search for information in the literature about this technique returns scarce results due to its relative newness in restorative dentistry and the limitation of clinical studies. This is the first case report of the injectable resin technique presented step by step in a patient with cleft lip and palate.

This clinical report aims to describe the use of the injectable fluid resin technique, with some modifications to the technique, for the re-anatomization of canines in a young patient with cleft lip and palate. Some parameters were observed during the execution of the technique, such as application time and marginal adaptation. In addition, restorations were performed on adjacent teeth (lateral incisors) using an incremental technique with conventional resins, which aided in the assessment of visual and photographic color stability and wear resistance between the two restorative approaches after one year of follow-up.

## 2. Case Presentation

A female patient, 21 years old with unilateral left cleft lip and palate, presented at the Dentistry Department of the Hospital for Rehabilitation of Craniofacial Anomalies at the University of São Paulo (HRAC/USP) with aesthetic complaints in the anterior upper region. After anamnesis and clinical examination, it was noted that teeth 13 and 23 had been re-anatomized into 12 and 22 after orthodontic treatment was completed when the patient was 16 years old (Figure 1). The patient received a bone graft in the region of the cleft with recombinant human Bone Morphogenetic Protein-2 (rh-BMP2) when she was 12 years old. The patient had agenesis of tooth 22 and tooth 12 had a short root. After the bone graft and during orthodontic treatment for correction, the patient had the option of having an implant in the region of teeth 12 and 22, but she and her parents wished to simplify the treatment by extracting tooth 12, undergoing mesialization of their teeth with orthodontic treatment, and maintaining group occlusion instead of canine guidance. It was observed that after 5 years of completed orthodontic treatment, the patient maintained a stable occlusion without recessions in the premolars. The resin on the “lateral” teeth was opaque with recurrence (small diastemas between 12 and 11 and on the distal of 22). Thus, the patient was dissatisfied with the color of her teeth and did not want to repeat orthodontic treatment. Therefore, treatment options were presented after initial case planning with photos and aesthetic evaluation. Restorative techniques were recommended and explained, but dental bleaching and a combination of restorative techniques were chosen, including the injectable composite resin technique to re-anatomize teeth 14 and 24 into 13 and 23 and replacement of the restorations of teeth 12 and 22 using the direct composite resin technique (Filtek XT Z350 B2B and B2E, 3M, Two Harbors, MN, USA) and closing diastemas.

First, an initial impression of the upper and lower arches was taken with alginate to make trays for at-home bleaching with 10% carbamide peroxide (Whiteness Perfect 10%, FGM, São Paulo, SP, Brazil) for 3 h a day for 30 days. With this step, the patient went from the initial color A3 on the Vita Classical scale to B1 (Figure 2). After the completion of bleaching, a new impression was made with condensation silicone to wax the teeth to be re-anatomized (Figure 3). Based on the case planning and wax-up, the patient chose not to undergo periodontal surgery and instead opted to improve the proportion of the gingival contour. Thus, it was decided to re-anatomize teeth 14 and 24 into 13 and 23 using the transparent matrix technique with injectable composite resin (Tetric N-Flow BL L, Ivoclar Vivadent, SP, Brazil).

For the injectable technique, the waxing model was replicated by molding the wax-up. Subsequently, a transparent plate made of ethylene/vinyl acetate copolymer (EVA) was used, which is a modification of the original protocol that uses a silicone matrix. However, no damage was observed from this adaptation. Therefore, the plate was taken to a vacuum laminator to obtain the transparent matrix. Later, small holes were made in the transparent matrix with high-speed diamond tips in the region of the buccal cusps of the premolars that would be re-anatomized to give access to the tip of the fluid resin to be injected as a transfer vehicle between the tooth and the resin (Figure 4).

Before the matrix was inserted into the oral cavity, a proper fit on the teeth was verified so that the fluid resin could be injected adequately. Prior to this, prophylaxis was performed using pumice stone paste to remove the biofilm and leave the dental surface clean. Subsequently, adjacent teeth were covered with polytetrafluoroethylene (PTFE) tape to prevent material adhesion to undesired dental surfaces and to allow optimal integration in the interproximal region during composite resin infiltration. Teeth 14 and 24 were etched with 37% phosphoric acid for 30 s, followed by a 30 s rinse, gentle drying with air jets to remove excess moisture, the application of the adhesive system (Prime & Bond 2.1; Denstiply, Sirona, SP, Brazil) (Figure 5), and 30 s of light curing (Radi Call SDI, Trevose, PA, USA). When the transparent matrix was positioned, fluid resin was injected to fill the space in tooth 14 between the tooth and the silicone matrix (Figure 6). When the space was filled with the material, light curing was performed for 60 s. The same procedure was repeated for tooth 24 (Figure 6). After the restorations were completed, the matrix was removed, the excess material was removed with a No. 12 surgical blade, and additional light curing was performed for 20 s.

A glycerin gel was also applied during restoration for final photopolymerization that inhibited the oxygen layer, thus preventing discoloration of the restoration. Occlusion was checked with 12 and 8 µm articulating paper (AccuFilm, Parkell, Long Island, NY, USA), verifying the absence of premature contacts and correct occlusal guidance (anterior and group guide). The composite resin colors to be used on teeth 13 and 23 were selected prior to the resin infiltration procedures on teeth 14 and 24. Teeth 13 and 23, which had already been re-anatomized into “12 and 22,” had the old composite resins carefully removed and replaced using the direct composite resin technique (Filtek XT Z350 B2B and B2E, 3M ESPE) (Figure 7).

The patient was satisfied with the restorative treatment, exhibiting a more harmonious and age-appropriate smile. They chose not to alter the shape or add resin to teeth 11 and 21, as their desired outcome had already been achieved (Figure 8).

After 1 year, the patient was called for a check-up, and new photos were taken after prophylaxis. The restorations made with the injectable resin technique did not show marginal discoloration or fracture, presenting similar results to the incremental composite resin technique, which also did not change (Figure 9). Therefore, no finishing or polishing was performed.

## 3. Discussion

In the literature, different restorative approaches have been proposed to perform aesthetic restorations. The present case describes the use of the injectable fluid resin technique with some modifications, which allowed for the re-anatomization of the upper premolars in canines in a young patient with cleft lip and palate, as well as a direct restorative approach with composite resin for teeth 12 and 22. 

The use of the injectable fluid resin technique was chosen for the premolars as there is evidence in the literature that new fluid composite resins with higher load values (61 to 71% by weight) offer better material adaptation to the cavity walls as well as to the posterior walls, with fewer failures compared to conventional resins. In addition, recent studies [8,9,30,31] show that higher-load fluid resins offer higher mechanical properties (flexural strength and flexural modulus), higher resilience, a high modulus of elasticity [30,31], wear resistance, and better polishing and shine than conventional universal resins [8,9]. Another advantage of this technique is the low cost and good aesthetic results, since the silicone matrix is a replica of the diagnostic wax-up. Thus, the position, shape, and contour are more faithful compared to other incremental techniques [10]. In addition, this technique provides the possibility of restoring one or more teeth, even in cases involving posterior teeth [32]. However, reports in the literature indicate that some care must be taken to avoid periodontal biological complications [10,25,29]. One study [10] indicated that to avoid invasion of the biological distance, the retraction cord should be placed in the gingival sulcus to prevent the fluid resin from flowing more than 0.5 mm; other authors [25,29] indicated that the transparent matrix can be cut following the gingival margin and at the sulcular level, thus obtaining better control of excess material and facilitating its removal. In this clinical case, the placement of a retraction cord in the gingival sulcus was chosen, thus avoiding infiltration of the fluid resin into the gingival sulcus. 

The treatment of a patient with cleft lip and palate should be guided by rehabilitation principles, such as physiology, stability, aesthetics, hygiene conditions, and individual expectations [33], which are important to the guidance and initiation of dental treatment. Anomalies of number and form in the teeth behind the cleft lip are common in patients with cleft lip and palate [13,14,20], and when trying to rehabilitate this area, it is common to re-anatomize teeth and sometimes not follow the principals of gold proportions in aesthetic rehabilitation. In this case report, the canines that were re-anatomized in the lateral incisors presented an unfavorable dental proportion in relation to the golden ratio. They ended up with a width close to the central teeth, which required the application of anatomical principles of illusion to create the appearance of narrower teeth. This technique involves altering the position of the vertical edges, favoring the principles of the golden ratio, which is considered one of the options to assist us in aesthetics. The injectable technique seems to be an interesting alternative in the restorative treatment of these patients considering situations where aesthetic treatment should be optimized in a specialized location, such as HRAC/USP. The hospital from this study is a world-class reference in the treatment of cleft lip and palate and hearing loss, was founded in the 1960s, and offers multidisciplinary treatment to numerous patients during the year. Thus, patients who come for dental treatment stay at HRAC for a few days depending on their needs. Analyzing the case report of this article, the technique used facilitated the dental session and promoted the result expected by the patient, without additional appointments.

The injectable technique is generally a simple treatment option for patients looking to improve aesthetics, especially in cases of small re-anatomizations. Therefore, randomized controlled clinical studies are required to demonstrate the long-term effectiveness of the technique. A recent study [34] related to the longevity of restorations, as well as the selection and use of restorative materials alone, indicates that the success of restorations will depend on a series of patient-related risk factors, such as age, the individual’s associated risks, parafunctional habits, and the size of the restoration, since the greater the amount of dental structure replaced by a polymeric composite, the greater the mechanical challenges imposed on the restoration. 

Thus, the authors indicate that the impact of photopolymerization on the longevity of restorations has been widely discussed in the literature [34,35], and there is enough evidence that the differences between resins from different manufacturers are not an important factor that influences the longevity of composite restorations [34,36,37]. Therefore, this study indicates that professionals are free to select materials based on handling preferences, color availability, technical aspects, and ease of polishing [34,36,37].

In this clinical case report, no differentiations were found between restorative techniques in relation to marginal discoloration and fracture resistance. Therefore, the clinic may have another clinical option for restorative treatment in the case of small re-anatomizations in young patients. However, the clinic should always consider patient-related risk factors such as age, lifestyle, and parafunctional habits, which play an important role in the success of restorations [34].

In addition, one of the limitations of case reports, although they can be useful to present interesting or unusual clinical findings, is that they should not be used as definitive evidence to support the effectiveness of treatment. However, they can be useful to suggest new research hypotheses that can be investigated in more rigorous clinical studies.

## 4. Conclusions

Smile rehabilitation in patients with cleft lip and palate should be associated with aesthetic and conservative techniques. Among the restorative treatment options, the use of the injectable fluid resin technique seems to be an interesting option for cases involving small re-anatomizations and requiring less chair time and good marginal adaptation. Furthermore, no color change or fracture was observed after one year of follow-up.

## Figures and Tables

**Figure 1 medicina-59-00849-f001:**
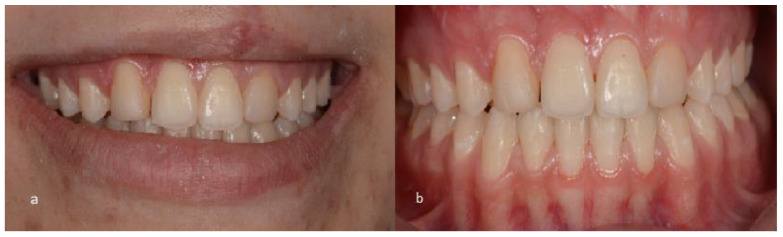
Initial photos of the 21-year-old patient. (**a**) Smile; (**b**) intraoral view of the teeth.

**Figure 2 medicina-59-00849-f002:**
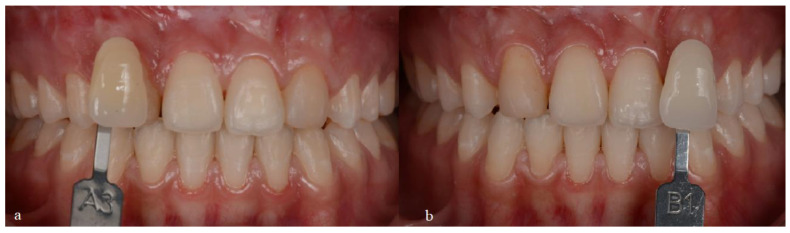
Photos of the bleaching treatment: (**a**) initial color (A3 on the Vita Classical scale); (**b**) final color (B1 on the Vita Classical scale).

**Figure 3 medicina-59-00849-f003:**
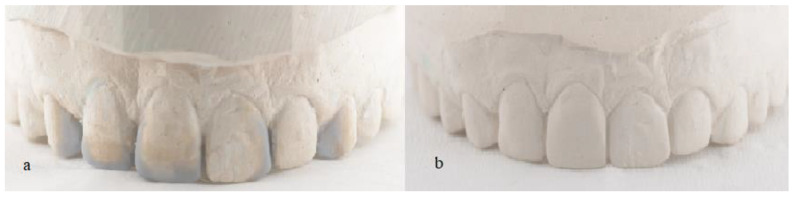
(**a**) Analog waxing; (**b**) copy of the wax-up for making the transparent matrix for treatment with the injectable composite resin technique.

**Figure 4 medicina-59-00849-f004:**
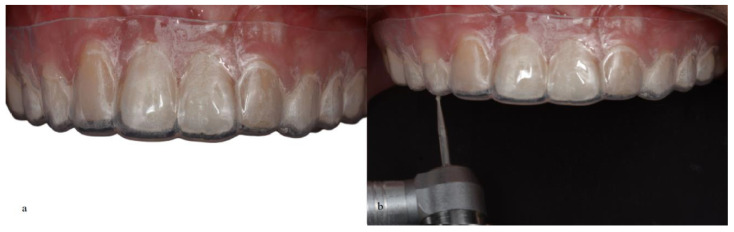
(**a**) Transparent matrix positioned over the patient’s teeth. (**b**) Drilling with diamond burs to open small holes in the transparent matrix.

**Figure 5 medicina-59-00849-f005:**
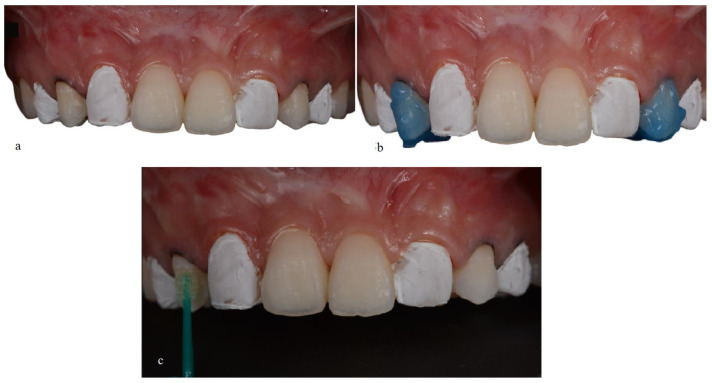
(**a**) Isolation of adjacent teeth with polytetrafluoroethylene tape and the placement of gingival retraction cords. (**b**) Phosphoric acid attack. (**c**) Application of the adhesive system.

**Figure 6 medicina-59-00849-f006:**
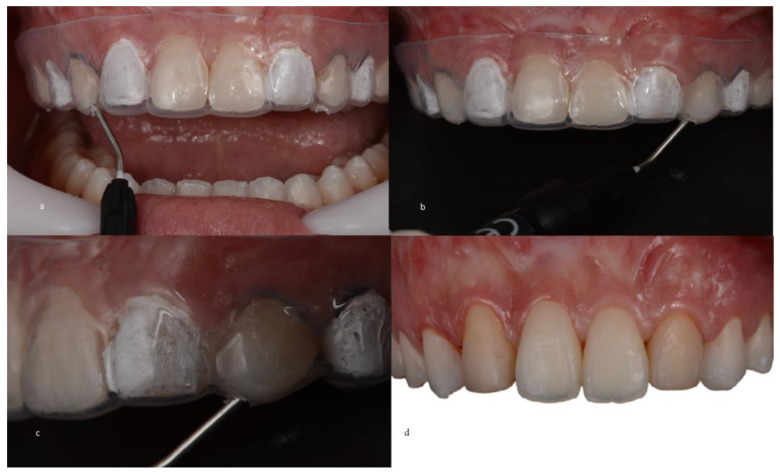
(**a**) Placement of the matrix on the teeth. (**b**) The tip of flowable composite resin inserted into holes. (**c**) Injection of the fluid composite into the transparent matrix, which was light-cured afterward. (**d**) Completed restorations after removal of the transparent matrix, Teflon tape, and gingival retractor cord.

**Figure 7 medicina-59-00849-f007:**
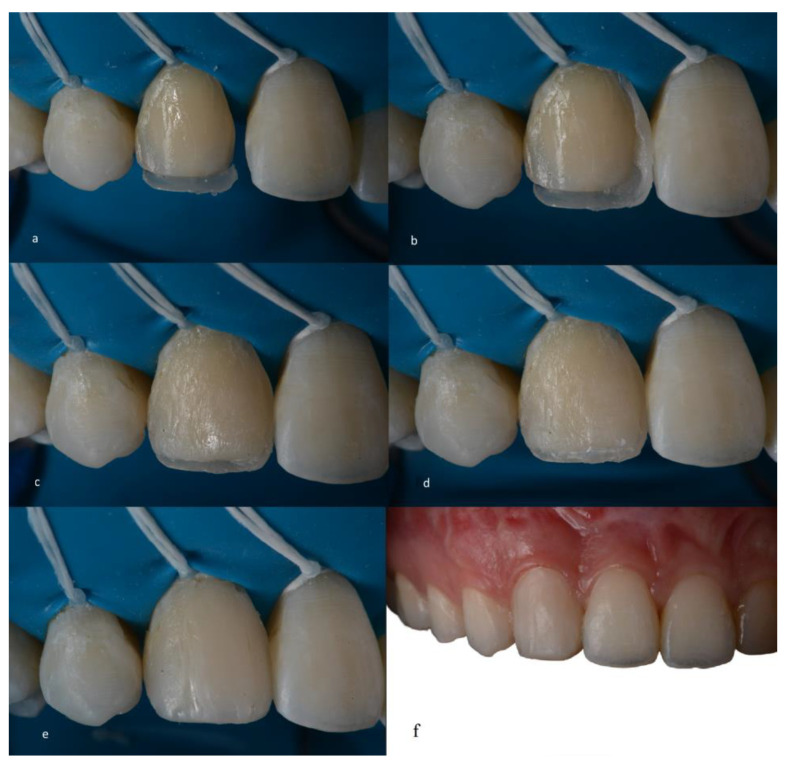
(**a**–**f**). Direct restoration sequence using the incremental technique with a conventional composite resin.

**Figure 8 medicina-59-00849-f008:**
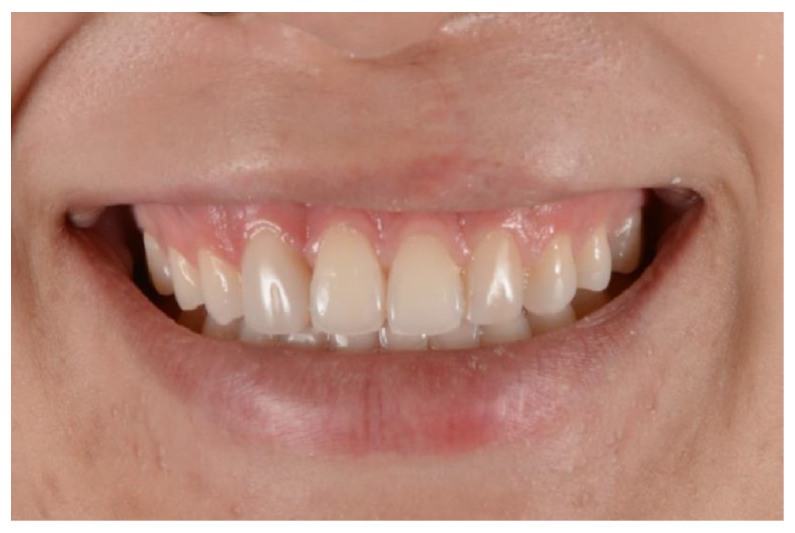
Natural smile. The final result of the restorations after finishing and polishing.

**Figure 9 medicina-59-00849-f009:**
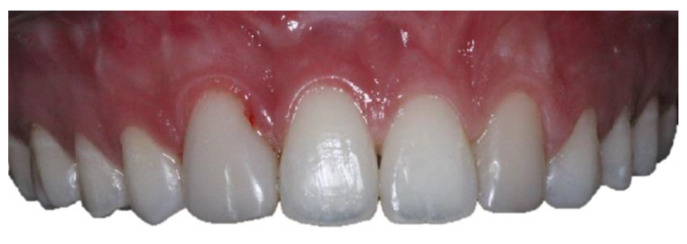
Photo after 1 year of restorations. Intraoral view after performing prophylaxis.

## Data Availability

Data can be obtained from the corresponding author, Kelly R. V. Villafuerte, PhD., upon reasonable request. Furthermore, the authors report that there is no financial interest in companies whose materials were included in this case report.

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
