# Peer review of "Injectable Resin Technique as a Restorative Alternative in a Cleft Lip and Palate Patient: A Case Report"

_medicina, 2023, doi:10.3390/medicina59050849_

Round 1
Reviewer 1 Report
This case report demonstrated the new restorative technique using injectable composite resin and a transparent matrix. The topic of this clinical report is interesting, and the treatment approach is important for the clinical issue. However, this manuscript should be revised for publication.
Before the restorative treatment, the author used home bleaching. What is the purpose of this bleaching, and is there any influence on the following restorations?
In this clinical case, they tried re-anatomizing teeth 14 and 24 into 13 and 23. The upper canine is a very important tooth as an occluded guide. However, the author did not evaluate the occlusion condition before and after restorations. Is there any data for bite condition? Please show the relevant occlusal data, and discuss it.
The anatomy of the upper canine and pre-molar is quite different. To re-anatomize these teeth, how to design occlusal and lateral surfaces? Did the author cut the tooth surface on both tooth surfaces? The journal readers could not understand the author’s clinical design for re-anatomizing teeth 14 and 24 into 13 and 23.
Did the author make some efforts to prevent the void using this flow composite resin?
Discussion: P8 lines 274-278: “Authors showed that the success of restorations depended on a series of risk factors related to patients,” In this case report, the author did not do any statistical analysis for the risk factor. Please rewrite this sentence.
This manuscript should be required linguistic and academic editing.
Author Response
Manuscript: Ref.: 2296976
Injectable resin technique as a restorative alternative in cleft lip and palate patient: A case report
Journal: Medicina
Dear reviewer,
Thank you very much for the time employed in our manuscript. All suggestions and comments made were helpful in improving our text.
The revised manuscript includes the changes considering the reviewers' comments (text in yellow) and a point-by-point response to each.
Questions:
This case report demonstrated the new restorative technique using injectable composite resin and a transparent matrix. The topic of this clinical report is interesting, and the treatment approach is important for the clinical issue. However, this manuscript should be revised for publication.
- Before the restorative treatment, the author used home bleaching. What is the purpose of this bleaching, and is there any influence on the following restorations?
Authors: Thank you very much for the comment. The home bleaching was done before the restorations, because the patient was unsatisfied with the color of her teeth, and did not have any influence on the following restorations, just on the color materials selected.
- In this clinical case, they tried re-anatomizing teeth 14 and 24 into 13 and 23. The upper canine is a very important tooth as an occluded guide. However, the author did not evaluate the occlusion condition before and after restorations. Is there any data for bite condition? Please show the relevant occlusal data, and discuss it.
Authors: This patient has a unilateral cleft and lip palate, and maybe in the region of the cleft it’s common to have changes in the teeth of this area in number and form like agenesis, hypomineralization, hypodontia, conoid tooth. So, this patient had agenesis of lateral 22 and the 12 has a short root. The patient had three options when was doing the orthodontic treatment: to do an implant in the region of tooth 22 and maintain the tooth 12(that was a little bit compromised because of its short root) or remove the teeth 12 and do two implants in the region of teeth 12 and 22 or remove the tooth 12 and with the orthodontic treatment close the spaces of teeth 12 and 22 with the mesialization of the others teeth. The patient with her parents decided to simplify the treatment and to not do implants, but they want to remove the tooth 12 to not has problems in the future. When it was done this kind of treatment, it was moving the oclusal conditions and losing the canine guide, but one thing that it is so important is to think in the group disocclusion. During the orthodontic treatment and when it was finished the group disocclusion was did. When the patient complain in the Operative Dentistry to improve her smile, she had finished the orthodontic treatment had 5 years, and it is possible to lock her stable occlusion without recession in premolars, that indicate an stable occlusion.
Line 103. as the patient had agenesis of tooth 22 and tooth 12 had a short root. During orthodontic treatment, the patient had the option of having an implant in the region of teeth 12 and 22, but she and her parents wanted to simplify the treatment by extracting tooth 12 and mesializing the teeth with orthodontic treatment, and maintaining group occlusion instead of canine guidance. It was observed that after 5 years of completed orthodontic treatment, the patient maintained a stable occlusion without recessions in the premolars.
- The anatomy of the upper canine and pre-molar is quite different. To re-anatomize these teeth, how to design occlusal and lateral surfaces? Did the author cut the tooth surface on both tooth surfaces? The journal readers could not understand the author’s clinical design for re-anatomizing teeth 14 and 24 into 13 and 23.
Authors: When it is necessary to re-anatomize one tooth sometimes It is important to cut the surface. In the canines, it was necessary to do a surface preparation, because canines had more volume (transverse edge), than the laterals did not. So, it was done surface preparation in canines, to improve the restorations and to re-constructed the vestibular surface of lateral teeth. The idea to restore the premolars in canines is just to enhance the canine tip, but without changing the occlusion.
- Did the author make some efforts to prevent the void using this flow composite resin?
Authors: To prevent void formations during the insertion of the flow composite resin, it was injectable with an applicator tip, and after the injection, before polymerization it was checked if it was some bubbles, and if it was presented, with an instrument, was tight until the bubble removed, and the polymerization was done.
- Discussion: P8 lines 274-278: “Authors showed that the success of restorations depended on a series of risk factors related to patients,” In this case report, the author did not do any statistical analysis for the risk factor. Please rewrite this sentence.
Authors: The sentence has been rewritten: Line 261. Although a recent study [35] related to the longevity of restorations, as well as the selection and use of restorative materials alone, indicate that the success of restorations will depend on a series of patient-related risk factors, such as age, caries risk, parafunctional habits, and size of the restoration, since the greater the amount of dental structure replaced by a polymeric composite, the greater the mechanical challenges imposed on the restoration.
- This manuscript must be required linguistic and scholarly editing.
Authors: Linguistic and academic editing was done by a native speaker
Regards,
The authors.

Reviewer 2 Report
Thank you.
My final suggestion is: The authors should try to concise the manuscript. As a case report, it's a large one.
Author Response
Manuscript: Ref.: 2296976
Injectable resin technique as a restorative alternative in cleft lip and palate patient: A case report
Journal: Medicina
Dear reviewer,
Thank you very much for the time employed in our manuscript. All suggestions and comments made were helpful in improving our text.
Reviewer: Thanks.
My final suggestion is: authors should try to concise the manuscript.
Authors: Thank you very much for the suggestion, we tried to make it as concise as possible, also figures considered unnecessary by some reviewers were removed.
Regards,
The authors.

Reviewer 3 Report
Dear Authors,
Here are some of my suggestions on how to improve your paper.
1. In the introduction, you should write more about the anomalies observed in cleft patients, like the problem with symmetries, microdontia etc. Also the Simonsart band should be mentioned as a great challenge to the regaining the function, please find more references of the below mentioned eg.:
Paradowska-Stolarz A, Mikulewicz M, DuĹ›-Ilnicka I. Current Concepts and Challenges in the Treatment of Cleft Lip and Palate Patients-A Comprehensive Review. J Pers Med. 2022 Dec 19;12(12):2089. doi: 10.3390/jpm12122089.
Ariawan D, Vitria EE, Sulistyani LD, et al. Prevalence of Simonart’s band in cleft children at a cleft center in Indonesia: A nine-year retrospective study. Dent Med Probl. 2022;59(4):509–515. doi:10.17219/dmp/145065
2. lines 82-84.... were the canines substituted first? There should be some information on the previous treatment of the patient (surgery?? - cleft closure + bone grafting? Any orthodontics??). No information on the type of cleft is provided, please add one - the reader is confused, in the photos we are not able to access what kind of cleft is that exactly?
3. Although the final result is pretty nice, please add the final outcome of teeth widths (it looks like the lateral "incisors" are not much narrower when compared to the centrals) - it should be also discussed in the discussion section, referring to the "golden proportion". Of course, there are some difficulties with this kind of restoration (they should be discussed further!). Maybe considering subjective and objective evaluation of the symmetry of maxillary incisors would be interesting here?
4. I am not pretty sure what Authors wanted to show us - the final outcome or the resins used? It should be highlighten. In this case I personally find the final outcome more interesting that the types of resins used (although, of course all should be mentioned). The problem is the Authors do not refer too much to the esthetic outcome, do not discuss it.
5. Did you use pontics to gain the gingival contour?
6. Should you use form "patients' wishes"? Or are they more "expectations"?
Thank you and good luck with further steps with this manuscript
Author Response
Manuscript: Ref.: 2296976
Injectable resin technique as a restorative alternative in cleft lip and palate patient: A case report
Journal: Medicina
Dear reviewer,
Thank you very much for the time employed in our manuscript. All suggestions and comments made were helpful in improving our text.
The revised manuscript includes the changes considering the reviewers' comments (text in yellow) and a point-by-point response to each.
Questions:
Here are some of my suggestions on how to improve your paper.
- In the introduction, you should write more about the anomalies observed in cleft patients, like the problem with symmetries, microdontia etc. Also the Simonsart band should be mentioned as a great challenge to the regaining the function, please find more references of the below mentioned eg.:
Paradowska-Stolarz A, Mikulewicz M, DuĹ›-Ilnicka I. Current Concepts and Challenges in the Treatment of Cleft Lip and Palate Patients-A Comprehensive Review. J Pers Med. 2022 Dec 19;12(12):2089. doi: 10.3390/jpm12122089.
Ariawan D, Vitria EE, Sulistyani LD, et al. Prevalence of Simonart’s band in cleft children at a cleft center in Indonesia: A nine-year retrospective study. Dent Med Probl. 2022;59(4):509–515. doi:10.17219/dmp/145065
Authors: Dear reviewer, thank you for your comments. The focus of the article is to describe the injectable resin technique as a restorative alternative for reanatomizing teeth in patients with cleft lip.
A paragraph has been added in the introduction, including references kindly sent:
Facial appearance is central to the psychosocial development of individuals with cleft lip, alveolus, and palate, who often exhibit characteristics of social introversion [17,18]. One of the main reasons may be dental anomalies, which have been shown to be higher in individuals with cleft compared to the normal population [19], is dental agenesis, supernumerary teeth, microdontic upper lateral incisors, ectopic eruption of teeth, delayed tooth eruption, and enamel hypoplasia, which are frequent in these patients [13,14,20,21]. Additionally, the presence of soft tissue bands connecting the cleft to the base of the nostril or the alveolar margin, known as Simonart's bands, is common [22,23]. These bands may contribute to a higher prevalence of distal upper lateral incisors to the cleft compared to patients without this condition [23]. Simonart's band could assist in the relationship in the upper arch, with a better prognosis in treatment, but it is not the only factor that will guarantee good resolution [23]. Therefore, these conditions result in long treatments carried out in hospitals and specialized places. Although dental treatments in most patients are usually completed between the ages of sixteen to eighteen years, some require additional aesthetic or functional treatment during adulthood [24].
- lines 82-84.... were the canines substituted first? There should be some information on the previous treatment of the patient (surgery?? - cleft closure + bone grafting? Any orthodontics??). No information on the type of cleft is provided, please add one - the reader is confused, in the photos we are not able to access what kind of cleft is that exactly?
Authors: Dear reviewer, tkanks for your attention. We try to explain more about the kind of cleft and lip palate the patient had, and the options in the treatment.
Line 103: The patient did a bone graft in the region of the cleft when she is 12 years with Recombinant human Bone Morphogenetic Protein-2 (rh-BMP2). The patient had agenesis of tooth 22 and tooth 12 had a short root. After bone graft, and during orthodontic treatment for correction, the patient had the option of having an implant in the region of teeth 12 and 22, but she and her parents wanted to simplify the treatment by extracting tooth 12 and mesializing the teeth with orthodontic treatment, and maintaining group occlusion instead of canine guidance. It was observed that after 5 years of completed orthodontic treatment, the patient maintained a stable occlusion without recessions in the premolars.
- Although the final result is pretty nice, please add the final outcome of teeth widths (it looks like the lateral "incisors" are not much narrower when compared to the centrals) - it should be also discussed in the discussion section, referring to the "golden proportion". Of course, there are some difficulties with this kind of restoration (they should be discussed further!). Maybe considering subjective and objective evaluation of the symmetry of maxillary incisors would be interesting here?
Authors: Thanks for the considerations, we try to discuss a little bit about.
Line 242., which are important to guide and initiate dental treatment. It is common in the patients with cleft lip and palate the anomalies of number and form in the teeth behind in the cleft lip [13,14,20], and try to rehabilitate this area, its common to re-anatomize teeth and sometimes to not follow the principals of gold proportions in aesthetic rehabilitation. In this case report, the canines that was re-anatomized in laterals, had almost the same width of the centrals, and it was necessary to use the anatomical principals of illusion, doing changes in the edges to had an illusion of a teeth more narrow.
- I am not pretty sure what Authors wanted to show us - the final outcome or the resins used? It should be highlighten. In this case I personally find the final outcome more interesting that the types of resins used (although, of course all should be mentioned). The problem is the Authors do not refer too much to the esthetic outcome, do not discuss it.
Authors: The esthetic outcome was added in the session “Case presentation” through the sentence: Line 187: The patient was satisfied with the restorative treatment, exhibiting a more harmoni-ous and age-appropriate smile. They chose not to alter the shape or add resin to teeth 11 and 21, as their desired outcome had already been achieved (Figure 8).
- Did you use pontics to gain the gingival contour?
Authors: Dear reviewer, we didn't quite understand the question. However, regarding the gingival contour, after planning and waxing the case, the patient did not want to undergo periodontal surgery to improve the proportion of the gingival contour. Therefore, it was decided to only re-anatomize the teeth.
- Should you use form "patients' wishes"? Or are they more "expectations"?
Authors: During treatment we try to resolve the wishes the patient with more information of the options she has. We could improve the teeth 11 e 21, but the wishes the patient was with the canines, that are transformed in laterals.
Regards,
The authors.

Reviewer 4 Report
“Injectable resin technique as a restorative alternative in cleft lip and palate patient: A case report” was submitted to Medicina.
The manuscript deals with an interesting issue; however, there are several concerns related to the study.
Abstract
Objective: There is no clear objective.
Results: The authors indicate “The clinical case report shows that the injectable technique offers simplicity and speed in the treatment..”. However, the authors do not indicate in the methodology the parameters used to report simplicity and speed.
Line 21. Shape is written twice.
Lines 21-22. “when comparing the two restorative techniques”. In the objectives, you must indicate that you want to make a comparison. The methods must describe the variables to be compared and the methodology used to do so.
Conclusion: The word "therefore" is not adequate to start a conclusion. The authors should focus on the results found.
How the following variables were measured: “less operator skill, chair time, and better marginal adaptation”. This case also did not evaluate risk factors. For this, more robust studies are required.
Keywords: Terms should be reviewed. Most of them are not MeSH terms.
Introduction
Lines 68-69. It is indicated that the literature is scarce. However, several studies are described in that paragraph. It is recommended to detail what the knowledge gap is.
Lines 72-75. The wording should be improved to better understand the additional objectives proposed. Moreover, the methodology must clearly indicate that a comparison is going to be made, indicating the variables to be compared and the form in which that comparison is measured.
M&M
Figures 1b and 2 need to be improved. The image of lips and retractors should be avoided.
Figure 3 lacks aesthetics. The surface of the teeth is defective, and many bubbles are visible on the models. They must be changed.
In figures 4a, 5a, 5b, 6d, 7f, and 9a, the black part must be removed.
Figures 8b and 9b are unnecessary.
Line 120. This information was already presented above.
Discussion
Line 252. More references should be included.
Line 282. More references should be included.
Lines 285-286. Considering the methodological aspects presented, this case cannot conclude these aspects.
Lines 288-290. Please add references.
This work has limitations that were not described. Some of them were mentioned above.
Conclusions
The authors should focus on the results found.
This manuscript needs to be edited by language services.
Author Response
Manuscript: Ref.: 2296976
Injectable resin technique as a restorative alternative in cleft lip and palate patient: A case report
Journal: Medicina
Dear reviewer,
Thank you very much for the time employed in our manuscript. All suggestions and comments made were helpful in improving our text.
The revised manuscript includes the changes considering the reviewers' comments (text in yellow) and a point-by-point response to each.
Questions:
Injectable resin technique as a restorative alternative in cleft lip and palate patient: A case report” was submitted to Medicina.
The manuscript deals with an interesting issue; however, there are several concerns related to the study.
Abstract
Authors: Thank you very much for the comment. The abstract was rewritten, following the reviewer’s considerations:
- Objective: There is no clear objective.
Authors: Was rewritten: The objective of this study is to present a case report in which the injectable composite resin technique was used as a restorative alternative for dental reanatomization in a patient with cleft lip and palate and aesthetic complaints.
- Results: The authors indicate “The clinical case report shows that the injectable technique offers simplicity and speed in the treatment..”. However, the authors do not indicate in the methodology the parameters used to report simplicity and speed.
Authors: In Materials and Methods was added: The treatment plan included the reanatomization of the maxillary premolars and canines using a flowable composite resin. This resin was injected and cured through a transparent matrix, which was a copy of the diagnostic wax-up model. Some parameters, such as application time, and marginal adaptation were also observed when performing the restorations.
Results: The clinical case report shows that the injectable technique was a simple and quick method for restoring the anatomy of teeth (shape and contour) in one session, since the injectable resin can be easily applied in interproximal areas and without the need to manually sculpt the resin. In this case, no clinical, visual, or photographic differences were found in marginal discoloration, color stability, and fracture/wear deterioration for the two restorative techniques after one year of follow-up.
- Line 21. Shape is written twice.
Authors: The second word “shape” was deleted.
- Lines 21-22. “when comparing the two restorative techniques”. In the objectives, you must indicate that you want to make a comparison. The methods must describe the variables to be compared and the methodology used to do so.
Authors: In this case, no clinical, visual, or photographic differences were found in marginal discoloration, color stability, and fracture/wear deterioration for the two restorative techniques after one year of follow-up.
- Conclusion: The word "therefore" is not adequate to start a conclusion. The authors should focus on the results found.
Authors: The word “therefore” was deleted: The professional may have another clinical option for restorative treatment in case of small re-anatomizations. In addition, the injectable technique seems to require less operator skill, chair time and better marginal adaptation in cases of small anatomical changes.
How the following variables were measured: “less operator skill, chair time, and better marginal adaptation”. This case also did not evaluate risk factors. For this, more robust studies are required.
Regarding the variables: Chair time, is the time the operator took to complete the procedure, whether it was in one or more sessions. In addition, chair time was also used as an indicator of skill level, whether or not it was quick to perform the restoration, as it did not require sculpting the resin. The marginal adaptation of the restoration was evaluated by observing the interface between the tooth and the restoration. In this case, the operator was able to place the resin precisely and create a restoration that sat well with the tooth.
- Keywords: Terms should be reviewed. Most of them are not MeSH terms.
- Lines 285-286. Considering the methodological aspects presented, this case cannot conclude these aspects.
Authors: The sentence was replaced to: Smile rehabilitation in patients with cleft lip and palate should be associated with aesthetic and conservative techniques. Among the restorative treatment options, the use of the injectable fluid resin technique seems to be an interesting option for cases of small reanatomizations.
- Lines 288-290. Please add references.
Authors: Was added. Demarco, F.F.; Cenci, M.S.; Montagner, A.F.; de Lima, V.P.; Correa, M.B.; Moraes, R.R.; Opdam, N.J.M. Longevity of composite restorations is definitely not only about materials. Dental materials: official publication of the Academy of Dental Materials 2022, doi:10.1016/j.dental.2022.11.009.
- This work has limitations that were not described. Some of them were mentioned above.
Authors: Line 280:The limitation was added. One of the limitations of case reports, although they can be useful to present interesting or unusual clinical findings, they should not be used as definitive evidence to support the effectiveness of treatment. However, they can be useful to suggest new research hypotheses that can be investigated in more rigorous clinical studies.
- Conclusions: The authors should focus on the results found.
Authors: The conclusion was replaced to: Smile rehabilitation in patients with cleft lip and palate should be associated with aesthetic and conservative techniques. Among the restorative treatment options, the use of the injectable fluid resin technique seems to be an interesting option for cases of small reanatomizations, requiring less chair time and good marginal adaptation. Furthermore, no color change or fracture was observed after one year of follow-up.
- This manuscript needs to be edited by language services.
Authors: Linguistic and academic editing was done by a native speaker
- This work has limitations that were not described. Some of them were mentioned above.
Authors:The limitation was added Line 280. One of the limitations of case reports, although they can be useful to present interesting or unusual clinical findings, they should not be used as definitive evidence to support the effectiveness of treatment. However, they can be useful to suggest new research hypotheses that can be investigated in more rigorous clinical studies.
Regards,
The authors.

Round 2
Reviewer 3 Report
Dear Authors,
thank you for improvements. There are still some issues I would like you to correct:
- line 106 - please use word "expected", "wished" or "agreed to the treatment plan that assumed..." not "wanted"
- line 249 - this sentence sounds unsuitable for the scientific research, it is more like "talking to the patient" - please change it into more scientiffic manner
- please, prepare references according to the journals scope.
Thank you
Author Response
Manuscript: Ref.: 2296976
Injectable resin technique as a restorative alternative in cleft lip and palate patient: A case report
Journal: Medicina
Dear reviewer,
Thank you very much again for all the suggestions and comments you made were helpful to improve our text.
The revised manuscript includes the changes considering the reviewers' comments (yellow text) and a point-by-point response for each.
Questions:
- line 106 - please use word "expected", "wished" or "agreed to the treatment plan that assumed..." not "wanted"
Authors: Dear reviewer, the word has been changed to “wished”
- line 249 - this sentence sounds unsuitable for the scientific research, it is more like "talking to the patient" - please change it into more scientiffic manner
Authors: The sentence has been rewritten to a more scientific manner
En este caso de relato, los caninos que fueron reanatomizados en los incisivos laterales presentaron una proporción dental desfavorable en relación con la proporción áurea. Terminaron con un ancho cercano a los dientes centrales, lo que requirió la aplicación de principios anatómicos de ilustración para crear la apariencia de dientes más estrechos. Esta técnica consiste en alterar la posición de los bordes verticales, favoreciendo los principios de la proporción áurea, que se considera una de las opciones para ayudarnos en la estética.
- Por favor, prepare referencias de acuerdo con el alcance de la revista.
Autores: Las referencias fueron colocadas de acuerdo con el alcance de la revista.
Saludos
Los autores.

Reviewer 4 Report
The authors made the recommended adjustments; therefore the publication of this manuscript is recommended.
Author Response
Manuscript: Ref.: 2296976
Injectable resin technique as a restorative alternative in cleft lip and palate patient: A case report
Journal: Medicina
The authors made the recommended adjustments; therefore the publication of this manuscript is recommended.
Thank you very much
Regards,
The authors.
